# Association of Foot Sole Sensibility with Quiet and Dynamic Body Balance in Morbidly Obese Women

**Jair Wesley Ferreira Bueno** [1] , **Daniel Boari Coelho** [1,2], **Caroline Ribeiro de Souza** [1] **and Luis Augusto Teixeira** [1,*]

[1] Human Motor Systems Laboratory, School of Physical Education and Sport, University of São Paulo, São Paulo 05508-030, Brazil; jwesleybueno@usp.br (J.W.F.B.); daniel.boari@ufabc.edu.br (D.B.C.); caroline.cahrs@gmail.com (C.R.d.S.)

[2] Biomedical Engineering, Federal University of ABC, São Bernardo do Campo 09606-045, Brazil

\* Correspondence: lateixei@usp.br; Tel.: +55-11-3091-2129

**Abstract:** An important health-related problem of obesity is reduced stance stability, leading to increased chance of falling. In the present experiment, we aimed to compare stability in quiet and in dynamic body balance between women with morbid obesity ($n$ = 13, body mass index [BMI] > 40 Kg/m$^2$, mean age = 38.85 years) and with healthy body weight (lean) ($n$ = 13; BMI $\leq$ 25 Kg/m$^2$, mean age = 37.62 years), evaluating the extent to which quiet and dynamic balance stability are associated with plantar sensibility. Quiet stance was evaluated in different visual and support base conditions. The dynamic task consisted of rhythmic flexion—extension movements at the hip and shoulder, manipulating vision availability. The plantar sensibility threshold was measured through application of monofilaments on the feet soles. The results showed that the morbidly obese, in comparison with the lean women, had higher plantar sensibility thresholds, and a reduced balance stability in quiet standing. Mediolateral stance stability on the malleable surface was strongly correlated with plantar sensibility in the obese women. Analysis of dynamic balance showed no effect of obesity and weaker correlations with plantar sensibility. Our results suggest that reduced plantar sensibility in morbidly obese women may underlie their diminished stance stability, while dynamic balance control seems to be unaffected by their reduced plantar sensibility.

**Keywords:** high obesity; plantar sensitivity; stance stability; body balance; dynamic balance





## 1. Introduction

Among anthropometric parameters, body weight has been revealed to be one of the strongest predictors of body balance stability. Data presented by Hue et al. [1] indicated that body weight accounts for over 50% of balance stability variance. Association between body weight and balance stability is particularly relevant for obese individuals. As indicated by evaluation of amplitude and velocity of feet soles center of pressure (CoP) sway in the anteroposterior (AP) and mediolateral (ML) directions, balance stability in a quiet upright stance is decreased in obese women as compared to lower weight groups [2–7]. Moreover, Dutil et al. [2] showed that obese, in comparison with lean individuals, presented an increased range of CoP sway when visual information was prevented, suggesting a greater dependence on vision for balance control. Association between body weight and balance stability can be thought to be critical in cases of extreme (morbid) obesity, when body mass index (BMI) is equal or higher than 40 Kg/m$^2$ [5], given the evidence that obesity is associated with a higher risk of having a fall [8,9]. Additionally, gender-related comparisons suggest that women may have their balance more importantly affected than men [3], requiring thus increased attention regarding obesity-related balance deficits.

Dynamic balance tasks may be even more challenging for obese individuals than keeping a quiet stance, as suggested by their increased difficulty in daily living tasks such as trunk bending, kneeling, stooping, and lifting or carrying bags [10]. Further support for this assumption has been presented by showing that the effect of obesity on dynamic

balance can be observed in regular reaching when standing, with obese individuals being featured by moving their trunk forward to perform the action while non-obese individuals predominantly move their reaching arm only [11]. In more challenging tasks, such as maximum forward trunk leaning for reaching, obese compared to lean young adults were shown to have decreased maximum reaching distance [4] and to take a longer time to complete the movement [12]. A reduced forward limit of balance stability has been shown to be characteristic of individuals with morbid obesity, but not in those with lower obesity grades [13]. This finding indicates the relevance of looking particularly into the effects of extreme obesity on dynamic balance stability.

Teasdale et al. [14] have hypothesized that one of the main factors leading to reduced balance stability in obese individuals is a lower sensibility of plantar mechanoreceptors as a result of continuous pressure exerted by body weight on the feet soles when standing. Analysis of quiet stance has shown that obesity leads to increased feet soles pressure mainly under the forefoot [15,16], midfoot lateral arch [16–18], and under the heels [16]. Assessing the association between body weight and plantar sensibility, Ajisafe et al. [19] found that obesity was inversely associated with feet sole sensibility (see also [20]). Conversely, recent results showed that BMI and plantar sensibility are correlated in lean but not in morbid obese women [21]. This finding suggests a limit (floor effect) for plantar sensibility decrease in extreme obesity. A previous investigation has evidenced the consequence of obesity-related low plantar sensibility on body balance by showing correlation between reduced plantar sensibility and increased postural sway in quiet standing [22]. Contrary to the assumption that poor feedback from the plantar mechanoreceptors in the obese individuals could be compensated for by other sensory sources, visual occlusion and disruption of vestibular information were shown to not affect differentially obese and lean individuals. From the aforementioned results, one could assume that very low plantar sensibility as a consequence of morbid obesity may be critical for quiet standing and possibly even more critical to dynamic balance stability. However, previous research has not addressed the potential association between feet sole sensibility and balance stability in quiet and dynamic tasks in morbid obese individuals.

In the present investigation, we aimed to compare body balance stability both in quiet standing and in dynamic trunk oscillation tasks between morbidly obese and lean women. To further understand potential factors leading to balance deficits associated with high levels of obesity, we assessed the extent to which plantar sensibility can account for quiet and dynamic balance stability. Given that reduced plantar sensibility in obese individuals may lead to intersensory compensation, with possible increased use of visual feedback [2], we tested the participants in the conditions of eyes open and closed. We also disrupted tactile feedback from the feet soles by comparing rigid and malleable support surfaces. Because of the reduced foot sole sensibility in the obese participants, these sensory manipulations were expected to affect this group to a greater extent than lean women. We hypothesized that morbidly obese in comparison with lean women have a decreased quiet (H1) and dynamic (H2) balance stability, and that balance stability is correlated with plantar sensibility (H3).

## 2. Materials and Methods

### 2.1. Participants

A convenience sample of women with morbid obesity (*n* = 13 (Sample size was defined based on Meng et al. [7])), BMI $\geq$ 40 Kg/m$^2$, mean age 38.85 years (*SD* = 8.09, range 27–55 years), or lean (*n* = 13), BMI $\leq$ 25Kg/m$^2$, mean age 37.62 years (*SD* = 7.10, range 27–53 years), were evaluated in this study. Table 1 presents descriptive anthropometric data for each group, in addition to the respective weekly amount of time dedicated to moderate and vigorous walking (as estimated through the short version of the International Physical Activity Questionnaire [23]). Women with morbid obesity were contacted from a waiting list for bariatric surgery of the Clinics Hospital (São Paulo, Brazil), while the lean participants were contacted in the University of São Paulo (Brazil) community. Inclusion

criteria were self-declared absence of neurologic diseases, traumatic injuries to the lower limbs, skin lesions on the feet soles or peripheral neuropathy. Participants provided written informed consent, as approved by the Ethics Committee of the University of São Paulo, in accordance with the principles laid down in the Declaration of Helsinki.

**Table 1.** Descriptive characteristics of participants.

| | **Lean** | **Obese** |
|---|---|---|
| Weight (Kg) | 56.02 (5.22) [51–68] | 124.72 (16.59) [100–157] |
| Height (cm) | 159.0 (5.52) [148–171] | 160.38 (8,92) [143–177] |
| BMI (Kg/m$^2$) | 22.17 (1.79) [20.14–24.84] | 48.39 (4.01) [43.09–57.30] |
| Waist circumference (cm) | 79.08 (6.65) [69–89] | 135.54 (10.63) [121–150] |
| Hip circumference (cm) | 94.15 (3.85) [90–102] | 144.08 (9.07) [128–165] |
| IPAQ (min) | 197.31 (144.03) [0–525] | 315.77 (231.63) [0–780] |

Averages, standard deviation in parenthesis, range in brackets. BMI: body mass index. IPAQ: International Physical Activity Questionnaire.

*2.2. Evaluations, Equipment and Procedures*

Evaluations were made in a single session. Upon arriving at the laboratory, participants rested sat for 15 min. Preliminarily, body weight was measured through a force platform (OR6-6, Advanced Mechanical Technology, Inc., Watertown, MA, USA); height, waist circumference (measured at the midpoint between the 12th rib and the iliac crest) and hip circumference (measured at the widest hip width) were measured through a fabric tape. For evaluation of feet soles' sensibility, participants remained sat facing the examiner with the evaluated foot comfortably supported on a chair, keeping the leg barely parallel to the ground and eyes closed. The opposite leg was maintained relaxed, with the knee flexed at approximately 90° with the foot resting on the floor. Foot sole sensibility was evaluated by means of Semmes–Weinstein monofilaments (SORRI, Inc., Bauru/SP, Brazil) [24,25]. It consists of 6 nylon filaments of the same length and different diameters, considering logarithmic values in the applied load: 0.05, 0.2, 2, 4, 10 and 300 g-force (gf) [26]. To estimate the sensibility thresholds, the monofilaments were pressed perpendicularly to the receptive field hotspot of the following regions of the plantar surface: third and fifth toes, and hallux; first, third and fifth metatarsal heads; internal and lateral arches of the midfoot; and medial point of the heel (see [27,28]). Magnitude of monofilament pressure on each plantar hotspot was progressively increased until the participant reported perception of the correct touched point. Evaluation was performed in both the right and left feet soles, with a sequence of evaluated hotspots pseudorandomized across participants (for a detailed description, see Bueno et al. [21]). For increased consistency of pressure applied through the monofilaments, a single researcher examined plantar sensibility in all participants.

Balance stability was assessed by using a force platform (AMTI OR6-6). For evaluation of upright stance, participants stood barefoot, keeping their heels 15 cm apart, feet oriented slightly outward (about 15° from the midline), and arms relaxed hanging beside the trunk. The task aim was to keep stance as still as possible for 30 s. This task was performed on two surfaces by two visual conditions: supported on the rigid surface of the force platform, and on a malleable surface given by a high density foam topping the force platform (AirPad® BalancePad®, dimensions: 50 cm × 41 cm × 6 cm); eyes open, gazing at a visual target at eyes height, 2 m away, and eyes closed, keeping the head in the same orientation as for the eye open condition. Each surface by vision condition was evaluated by means of three consecutive trials spaced by 15 s of rest, and 1 min of seating rest every three trials. Sequences of testing conditions were alternated within the group across participants.

For the ensuing evaluation of dynamic balance, participants stood on the force platform with their feet at the same positions as that used for upright stance testing. For kinematic analysis of trunk movements, a spherical reflective marker (14 mm diameter) was attached to the right sided acromion. This marker was tracked through six optoelectronic cameras (Vicon Nexus, MX3+). The task consisted of cyclic movements

of hip flexion-extension, bending the trunk forward about 30 degrees in reference to the absolute vertical orientation, simultaneously with shoulders' flexion-extension movements, raising the arms at about the horizontal level (during hip flexion) and moving them back beside the trunk (during hip extension). During arm raising, the hands aimed at a spatial reference positioned at the individual shoulder height at a distance equivalent to 1.4 of the participant's arm length (Figure 1). In each trial, these movements were performed during 15 s, at a frequency of 0.5 Hz, paced by means of an electronic metronome (BOSS brand, model DB-60). This task was created for the purposes of the current experiment, aiming at inducing cyclic large movements of body segments standardized in amplitude and rhythm. This dynamic task was performed with the eyes open and closed. Performance on this task was evaluated through three consecutive trials for each visual condition (following two familiarization trials), and an alternating sequence of visual conditions across participants. We provided rest intervals of 15 s between trials within a visual condition, and of 1 min seating rest between trial blocks for each visual condition.

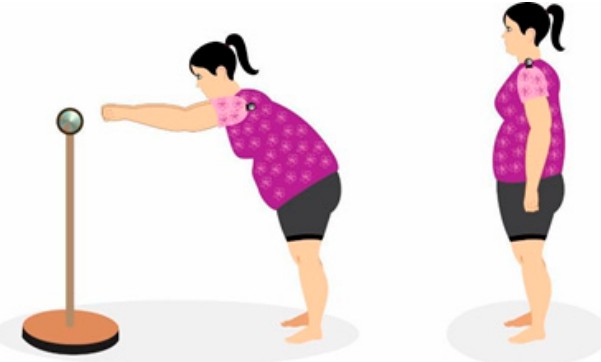

**Figure 1.** Representation of the dynamic task, with rhythmic oscillation between the two depicted body postures, with the top of the vertical shaft serving as a spatial reference for trunk bending and arm raising movements.

### 2.3. Data Collection and Analysis

Data sampling frequency was set at 200 Hz for both ground reaction forces and kinematics. Following preliminary visual inspection of individual trial signals, raw data were processed through MATLAB (MathWorks, Inc., Natick, MA 01760-2098, USA) routines. Kinematic and ground reaction force data were digitally low-pass filtered through a fourth-order zero lag Butterworth filter, with a cutoff frequency of 10 Hz. The following dependent variables were analyzed: root mean square (RMS) of center of pressure (CoP) displacement separately for the AP and ML directions; RMS of AP trunk oscillations (based on the shoulder marker) displacement, and mean frequency across movement cycles in the dynamic task; and scores for foot sole sensibility. For statistical analysis, monofilament logarithmic loading values were transformed into scores ranging from 1 to 6, from the thinnest (the highest sensibility) to the thickest (the lowest sensibility), respectively. Sensibility analysis was based on a global score given by the grand mean across foot sole regions and between the two feet (cf. [21], for a detailed analysis separated by plantar hotspots). Analysis of CoP displacement was based on individual averages of the three trials in each task.

As a prerequisite for parametric analysis, the Shapiro–Wilk test for CoP and trunk oscillation data showed normal distribution. Analysis of stability in the quiet balance task was made through three-way 2 (group: lean X obese) X 2 (surface: rigid X malleable) X 2 (vision: eyes open x eyes closed) ANOVAs for repeated measures on the last two factors, and analysis of the dynamic balance task was made through two-way 2 (group: lean X obese) X 2 (vision: eyes open X eyes closed) ANOVAs for repeated measures on the last factor. Post hoc comparisons were performed through Newman-Keuls procedures. Significant effects ($p < 0.05$) are reported only, accompanied by the respective effect

sizes given by partial eta squared ($\eta_p{}^2$). Spearman's rho ($r_s$) coefficients were used to analyze correlations between plantar sensibility scores and CoP RMS. Classification of correlation strength was adopted as it follows: very strong ≥ 0.8, strong ≥ 0.6 and <0.8, moderate ≥ 0.4 and <0.6, and weak < 0.4 (after [29]). Statistical analysis was performed using the Statistica software (version 7.0, Statsoft, Tulsa, OK, USA). The full dataset is available as Supplementary Material.

## 3. Results

### 3.1. Plantar Sensibility

Comparison of plantar sensibility scores between groups showed significantly higher scores (decreased sensibility) for the obese (*Med* = 2.83, first-third quartiles = 2.67–3.22) than the lean (*Med* = 2.05, first-third quartiles = 1.78–2.22) group, $Z$ = 3.79, $p$ < 0.01.

### 3.2. Quiet Stance

Anteroposterior CoP sway. Analysis of AP CoP sway indicated significant main effects of group, $F(1, 24)$ = 14.53, $p$ < 0.01, $\eta_p{}^2$ = 0.38; surface, $F(1, 24)$ = 554.21, $p$ < 0.01, $\eta_p{}^2$ = 0.96; and vision, $F(1, 24)$ = 60.14, $p$ < 0.01, $\eta_p{}^2$ = 0.71. The effect of group was due to higher values for the obese than the lean group; the effect of surface was due to higher values for the malleable than the rigid support surface; and the effect of vision was due to higher values for eyes closed than open (Figure 2A,B).

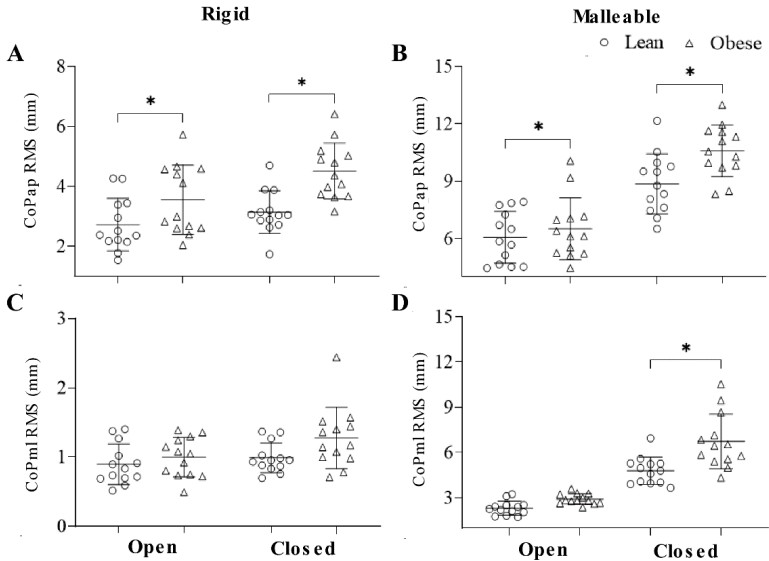

**Figure 2.** Individual data associated with group means and standard deviations (horizontal bars) of CoP RMS in the AP and ML directions for the rigid (**A**,**C**) and malleable (**B**,**D**) surfaces as a function of visual condition (eyes open versus closed); asterisks indicate statistically significant differences between the obese and lean groups.

Mediolateral CoP sway. Analysis of ML CoP sway indicated a significant group X surface X vision interaction, $F(1, 24)$ = 6.02, $p$ = 0.02, $\eta_p{}^2$ = 0.20. Post hoc comparisons indicated that the malleable in comparison with the rigid surface led to increased amplitude of ML sway in both groups. For the malleable but not rigid surface, eyes closed led to increased values than eyes open in both the obese and lean groups. The single between-group difference was found in the condition of eyes closed on the malleable surface, with higher values for the obese group (Figure 2C,D).

Spearman correlation analysis between plantar sensibility and CoP sway amplitude in the AP and ML directions is presented in Table 2, separately for each group and for the analysis including all participants. Results indicated that plantar sensibility was more clearly associated with balance stability in the ML direction, as the largest number of significant

correlations (eyes closed/rigid surface, and both visual conditions on the malleable surface, considering all participants) and the highest correlation values ($r = 0.72$ and $0.75$ [both strong], $r^2 = 0.52$ and $0.56$, respectively for eyes open and closed on the malleable surface) were found in that direction of CoP sway. For the obese women, in particular, analysis showed that on the malleable surface plantar sensibility was significantly correlated with balance stability in both eyes open ($r = 0.63$ [strong], $r^2 = 40$) and eyes closed ($r = 0.69$ [strong], $r^2 = 48$) visual conditions. When standing with the eyes closed, the analysis of all participants showed significant correlations not only in the ML but also in the AP direction in both support surfaces (rigid: $r = 0.48$ [moderate], $r^2 = 0.23$; malleable: $r = 0.44$ [moderate], $r^2 = 0.19$). A scatterplot representing the highest correlations on the malleable surface is shown in Figure 3 (panels A–B).

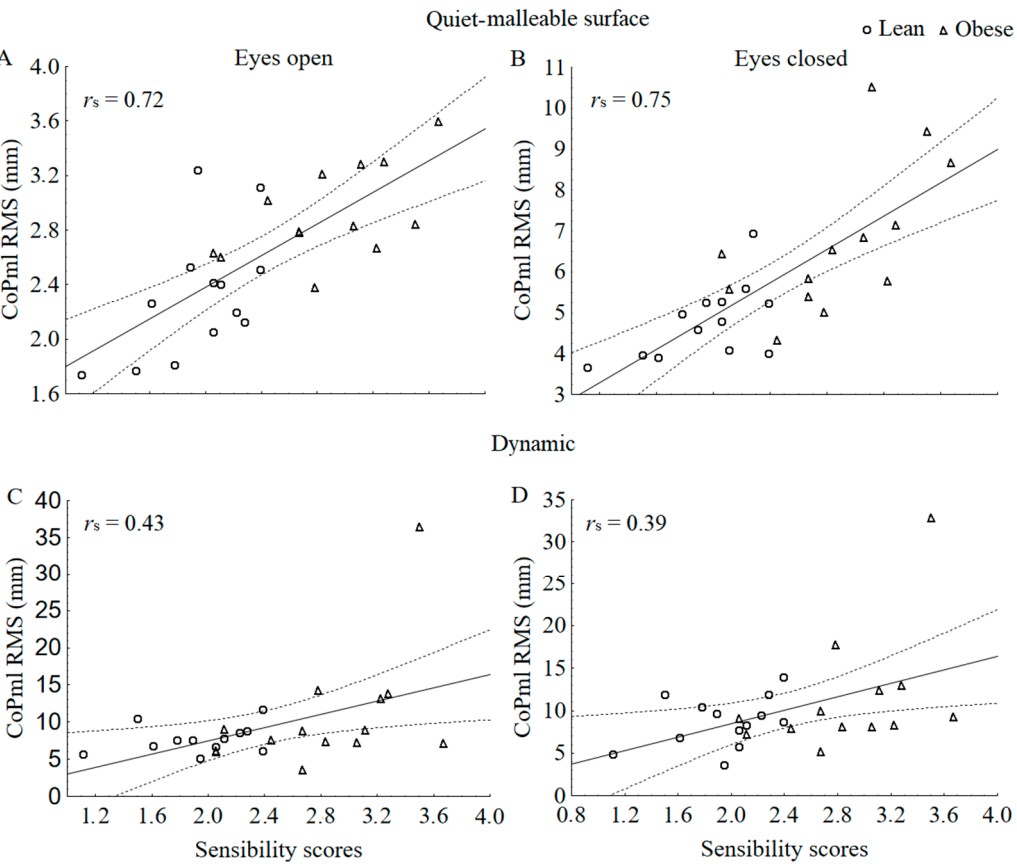

**Figure 3.** Scatterplot for the highest correlations between the plantar sensibility score and RMS of ML CoP sway on the malleable surface in quiet stance (upper panels, **A,B**) and in the dynamic task (lower panels, **C,D**) in the conditions of eyes open (left-sided panels, **A,C**) and closed (right-sided panels, **B,D**). Obese participants are represented by triangles and lean participants by circles. The dotted lines represent the confidence intervals.

**Table 2.** Correlation between plantar sensibility and CoP sway in quiet stance.

| | | Lean | | Obese | | All | |
|---|---|---|---|---|---|---|---|
| | | $r_s$ | $p$ | $r_s$ | $p$ | $r_s$ | $p$ |
| **Rigid** | | | | | | | |
| $CoP_{ap}$ RMS | open | 0.27 | 0.37 | −0.44 | 0.13 | 0.27 | 0.19 |
| (mm) | closed | 0.12 | 0.69 | −0.33 | 0.27 | 0.48 | 0.01 |
| $CoP_{ml}$ RMS | open | 0.72 | <0.01 | −0.37 | 0.21 | 0.31 | 0.13 |
| (mm) | closed | 0.68 | 0.01 | 0.22 | 0.48 | 0.61 | <0.01 |
| **Malleable** | | | | | | | |
| $CoP_{ap}$ RMS | open | −0.37 | 0.21 | −0.41 | 0.17 | −0.17 | 0.40 |
| (mm) | closed | 0.09 | 0.76 | 0.04 | 0.90 | 0.44 | 0.03 |
| $CoP_{ml}$ RMS | open | 0.52 | 0.07 | 0.63 | 0.02 | 0.72 | <0.01 |
| (mm) | closed | 0.57 | 0.04 | 0.69 | <0.01 | 0.75 | <0.01 |

### 3.3. Dynamic Balance

In Table 3 is presented results from kinematic analysis of AP trunk oscillations, movement frequency and amplitude (upper rows), and RMS of CoP sway in the AP and ML directions (lower rows). Analysis of trunk movement frequency indicated neither significant group nor vision related effects ($F$ values < 3.10, $p$ values > 0.09). For amplitude of trunk movements, the results showed a significant main effect of vision, $F(1, 24) = 5.67$, $p = 0.03$. This effect was due to lower values in the of eyes closed ($M = 71.80$ mm, $SD = 14.93$) than eyes open ($M = 77.95$ mm, $SD = 21.38$) condition. Analysis of CoP sway amplitude in the AP and ML directions showed no significant effects related either to group or vision ($F$ values < 3.40, $p$ values > 0.08).

**Table 3.** Trunk kinematic and CoP sway amplitude.

| | Lean | | Obese | | | |
|---|---|---|---|---|---|---|
| | **Open** | **Closed** | **Open** | **Closed** | **Factor** | $p$ |
| Trunk | 0.50 | 0.50 | 0.49 | 0.50 | group | 0.09 |
| frequency (Hz) | (0.01) | (0.01) | (0.02) | (0.02) | vision | 0.40 |
| Trunk | 73.73 | 69.07 | 82.16 | 74.53 | group | 0.31 |
| amplitude (mm) | (14.10) | (13.95) | (26.73) | (15.92) | vision | 0.03 |
| $CoP_{ap}$ RMS | 20.39 | 20.24 | 18.73 | 18.07 | group | 0.52 |
| (mm) | (3.97) | (4.47) | (12.44) | (7.75) | vision | 0.70 |
| $CoP_{ml}$ RMS | 7.62 | 8.69 | 11.03 | 11.47 | group | 0.18 |
| (mm) | (1.90) | (2.99) | (8.26) | (7.16) | vision | 0.08 |

Means, standard deviations (in parenthesis) and $p$ values for trunk and CoP oscillations, comparing the groups in the conditions of eyes open and closed in the anteroposterior (ap) and mediolateral (ml) directions.

Analysis of correlation between plantar sensibility and CoP sway in the dynamic balance task showed significant values for ML sway in both eyes open ($r = 0.43$ (moderate), $r^2 = 0.18$, $p = 0.03$, Figure 3C) and eyes closed ($r = 0.39$ [weak], $r^2 = 0.15$, $p = 0.05$, Figure 3D) conditions. As Figure 3C,D denotes, however, an outlier with high amplitude of CoP sway affected the correlation coefficients. In an extra analysis excluding this outlier, correlation coefficients fell below significance: $r = 0.38$ (weak, $r^2 = 0.14$, $p = 0.06$) and $r = 0.32$ (weak, $r^2 = 0.10$, $p = 0.12$), respectively for eyes open and closed. The whole correlation analysis is presented in Table 4.

**Table 4.** Spearman correlation coefficients ($r_s$) between plantar sensibility and CoP RMS in the dynamic task.

| | | Lean | | Obese | | All | |
|---|---|---|---|---|---|---|---|
| | | $r_s$ | $p$ | $r_s$ | $p$ | $r_s$ | $p$ |
| CoP$_{ap}$ (mm) | open | 0.44 | 0.13 | 0.11 | 0.73 | −0.18 | 0.37 |
| | closed | 0.22 | 0.47 | 0.24 | 0.42 | −0.17 | 0.42 |
| CoP$_{ml}$ (mm) | open | 0.30 | 0.32 | 0.35 | 0.24 | 0.43 | 0.03 |
| | closed | 0.31 | 0.30 | 0.54 | 0.06 | 0.39 | 0.05 |

Results for the anteroposterior (ap) and mediolateral (ml) directions in the conditions of eyes open and closed. Strength classification ($r$): very strong $\geq$ 0.8, strong $\geq$ 0.6 and <0.8, moderate $\geq$ 0.4 and <0.6, and weak < 0.4 (after Kutner et al. [29]).

## 4. Discussion

The current study was conducted with the purpose of comparing morbidly obese and lean women in tasks requiring quiet stance and dynamic balance stability, in addition to estimating the extent to which balance stability can be accounted for by plantar sensibility. Quiet stance analysis revealed that morbid obesity led to increased amplitude of body sway, mainly in the AP direction, while mediolateral sway was significantly increased in the obese women only in the more challenging condition of no vision when standing on the malleable surface. Surprisingly, we found no obesity-related effects on dynamic balance stability, even in the more challenging condition of no vision. Correlation analysis indicated that quiet stance control is more dependent on plantar sensibility than dynamic balance, with greater correlation coefficients observed in the former. Additionally, plantar sensibility was predominantly associated with mediolateral stability, and was more relevant for balance control in the challenging conditions of no vision and a malleable surface.

### 4.1. Quiet Balance

Our finding of a lower balance stability in quiet standing in the morbidly obese as compared to lean women is in agreement with previous findings in individuals with lower obesity grades [2–7], providing support for our first hypothesis (H1). A point worth noting in these results was that in the AP sway, the obese had significantly higher values than the lean women in all experimental conditions, while in the mediolateral direction lower balance stability in the obese women was found only in the most challenging condition represented by no vision supported on a malleable surface. Based on decreased plantar sensibility in obese individuals [19–21], one could expect greater dependence on vision for balance control (cf. [2]). A trend supporting this expectancy can be seen in both AP and ML directions of CoP sway in quiet standing, with larger descriptive mean differences between groups under no vision as compared to full vision (Figure 2). The statistically significant difference between groups in ML sway only on the malleable surface in the no vision condition suggests that the deficit of plantar sensibility in the obese women can be compensated for by an up-weighting of the use of visual feedback for balance control when this sensory information is available, while sensory information from the feet soles can be thought to be down-weighted (cf. [30,31]).

More evident decreased balance stability in the AP direction in obese women may be related to biomechanical and sensory factors. An increased inertial moment in the AP body sway due to the large body weight represents a mechanical constraint in plantar flexion actions for stance stabilization. In individuals with a high accumulation of fat in the abdominal region, CoP is shifted forward, leading to more anterior CoP positioning and a greater gravitational torque at the ankles [8,32]. In the sensory domain, recent results have shown that morbidly obese women have some of the most expressive lower plantar sensibilities under the fifth and third metatarsal heads, and under the heels [21]. Such reduced plantar sensibility from mechanoreceptors in these regions of the feet soles, associated with anteriorly displaced center of mass [8,32], may explain the lower stability

of the obese women in the AP as compared to the ML direction of balance regulation. Decreased quiet stance stability in obese individuals has been shown to be associated with enlarged use of attentional resources to keep balance control [33], and with an increased risk of falling [8,9], having thus potential consequences for daily living activities.

### 4.2. Dynamic Balance

Considering previous reports of increased difficulty to perform daily living actions requiring trunk and limb movements [4,10], and sensory deficits from the plantar mechanoreceptors [19–21], we hypothesized a decreased dynamic balance stability in morbidly obese women (H2). Our results refuted this hypothesis by showing a lack of significant differences between the obese and lean women when performing the dynamic task. A further point to be noted is that equivalent performance between the obese and lean groups was found also under no visual information, a condition requiring balance to be controlled from the available sources of sensory feedback, including information from the feet soles' mechanoreceptors. Divergent from evidence of relevance of sensory information from mechanoreceptors in the feet soles for quiet stance control [34–37], our results suggest that dynamic balance stability is less dependent on that feedback source. A possible explanation for this result is sensory re-weighting, with a flexible modulation in the central nervous system of the weights given to each sensory system based on information reliability for balance control [30,31]. In our dynamic task, we propose the interpretation that sensory information from the feet soles is less reliable for signaling the large amplitudes of body oscillation, leading to CoP sway of about 5–6 times higher (about 18–20 mm) than that observed in quiet standing (about 3–4 mm). Visual information during voluntary oscillatory head movements might be of reduced relevance for dynamic balance control as well, given the fundamental role of this sensory source of providing information on head stabilization in space [38,39]. On the other hand, behavioral [40] and brain imaging [41] studies on body balance have indicated that the contribution of the vestibular system increases when the visual information is unavailable. Considering that trunk oscillation frequency was voluntarily controlled, and then predictable, anticipatory feedforward mechanisms are thought to interact with feedback control to stabilize body balance [42–44]. Anticipatory control has been proposed to be mediated by vestibular-driven signals used to update an internal model formed on the basis of the predicted consequences of planned body movements [45]. Based on this conceptualization, the predicted consequences of upcoming voluntary trunk and limb movements to stance stability might lead to a reduced requirement of feedback from the feet soles and vision, while up-weighting proprioceptive information signaling body motion.

### 4.3. Plantar Sensibility and Balance

Previous results in individuals with lower grades of obesity have shown a weak to moderate correlation strength between plantar sensibility and balance stability in quiet standing [22]. Our results have revealed that such a correlation in morbidly obese women was strong, particularly when standing quietly on the malleable surface (eyes open $r = 0.63$, eyes closed $r = 0.69$), while only non-significant low to moderate correlation coefficients were observed for dynamic balance. These results, then, provided partial corroboration of the hypothesis of association between plantar sensibility and balance stability (H3). In quiet standing, the higher correlation strength in comparison with Wu and Madigan [22] data suggests that stance stability is further impaired as plantar sensibility is diminished due to the greater body weight in morbid obesity. This interpretation should be qualified, given that significant correlations in quiet stance were found in specific experimental conditions. It is worth noting that all strong correlations were found in the ML but not in the AP direction of CoP sway. This finding suggests that plantar sensibility can account for a large portion of mediolateral stability in quiet standing. Individual group analysis showed that this association was observed for both the obese and lean groups. The fact that some of the most consistent correlations between groups occurred when standing on

the malleable surface suggests that sensory information from the feet soles is more relevant for ML balance control in one of the most challenging conditions for balance stability (cf. Figure 2 panels C–D). This was particularly the case for obese women, with the highest correlation coefficients observed when standing on the malleable surface.

Previous results have provided clues for interpreting the finding that low foot sole sensibility in the obese women was associated with decreased balance stability (cf. Figure 3, panels A–B). Assessing foot sole sensibility thresholds to sensory stimulation in obese individuals, Miscio [46] showed that high sensory thresholds in those individuals were associated with a peripheral neural deficit represented by decreased sensory action potential amplitudes. Additionally, Lhomond [47] compared cortical responses in individuals with regular weight between the situations of supporting their own body weight against instantaneous increment of body weight by wearing a loaded vest. Results showed that the instantaneous body weight increment led to a fast rise in the sensibility threshold to foot sole stimulation in parallel with diminished cortical activation for early somatosensory evoked potentials. Those changes in somatosensory cortex activation by the added mass to the body seems to have disrupted early sensory transmission and subsequent sensibility-related processing of afferent feedback from the feet soles' mechanoreceptors. From these results, it is plausible that peripheral deficits of decreased sensory action potential amplitudes [46] and central deficits of processing sensory signals from the feet soles [47] can explain an important part of the decreased quiet stance stability in individuals with high levels of obesity. On the other hand, low to moderate correlations between plantar sensibility and dynamic balance stability is consistent with our interpretation (previous section) that sensory feedback from the feet soles is down-weighted when producing voluntary limb/trunk movements.

## 5. Conclusions, Applications and Limitations

Results from this investigation lead to the conclusion that morbidly obese, in comparison with lean, women are characterized by a low balance stability in quiet standing, while no obesity-related deficits were found in the dynamic balance. In the morbidly obese women, mediolateral balance stability when standing quietly on a malleable surface, but not in dynamic balance, was shown to be strongly correlated with sensibility thresholds on the feet soles. Our results suggest that reduced plantar sensibility in morbidly obese women underlies, to a great extent, their reduced quiet stance stability, whereas sensory feedback from the mechanoreceptors in the feet soles seems to be of lower relevance for the regulation of dynamic balance stability. As a clinical application, we suggest that balance training for morbidly obese women should prioritize improving body stability in the mediolateral direction particularly in the condition of absence of visual information while standing on malleable surfaces. The main limitations of this study were a lack of overweight and low-moderate obese participants to evaluate in a more comprehensive way the relationship between balance stability and plantar sensibility as a function of body weight and the limited numbers of participants potentially affecting the sensibility of statistical analyses.

**Supplementary Materials:** The following are available online at https://www.mdpi.com/article/10.3390/biomechanics1030028/s1. The full dataset is available as supplementary material.

**Author Contributions:** J.W.F.B., conceptualization, methodology, investigation, analysis, writing—original draft preparation; D.B.C., data curation, analysis, investigation; C.R.d.S., investigation, data processing; L.A.T., conceptualization, funding acquisition, supervision, writing, review and editing. All authors have read and agreed to the published version of the manuscript.

**Funding:** This work was supported by the Brazilian Council of Science and Technology (CNPq, grant number 306323/2019-2 [LAT]), and Coordination for the Improvement of Higher Level Personnel (CAPES, grant number 88882.327711/2019-01 [JWFB]).

**Informed Consent Statement:** Participants provided written informed consent.

**Data Availability Statement:** Full data set will be available as Supplementary Material.

**Acknowledgments:** The authors are thankful to Carla Rinaldin for the artwork of Figure 1.

**Conflicts of Interest:** The authors declare no conflict of interest. The funders had no role in the design of the study; in the collection, analyses, or interpretation of data; in the writing of the manuscript, or in the decision to publish the results.

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
