# Peer review of "Association of Foot Sole Sensibility with Quiet and Dynamic Body Balance in Morbidly Obese Women"

_2673-7078, doi:10.3390/biomechanics1030028_

Round 1
Reviewer 1 Report
While the authors make an interesting proposal, 3 fundamental aspects must be addressed before an in-depth review.
First, it is essential to incorporate a sample size calculation into the manuscript and duly justify why they use 13 women in each group.
Second, it must be justified because they incorporate open and closed eyes together in the ANOVA, and do not analyze separately as independent conditions. The vision factor must be explained and justified. In stability, there are specific tests with either open or closed eyes (e.g. emery test). If it is not able to be justified, it should be treated as two independent conditions and use t-test.
Finally, the clinical applications of the findings are not clearly described. Without this section and clear applications, the findings are irrelevant.
Author Response
Please, see the attachment.

Reviewer 2 Report
Review for Paper titled “Association of foot sole sensibility with quiet and dynamic body balance in morbidly obese women”
Abstract: Please edit the spacing between lines 17 and 18.
Method:
2.1 Participants: How was the sample size estimated?
2.2 Equipment: Please mention the company name/Country of the force plate(AMTI).
2.3 Data collection: Math works company detail.
Results: Line 218-224. The sentences need restructuring. It is hard to understand.
Provide the Limitation of the study.
All figures are required in higher resolution.
Please add units in each table for the measured variable.
Author Response
Please, see the attachment.

Round 2
Reviewer 1 Report
The authors have satisfactorily answered my questions.